# Effect of Phenolics from *Aeonium arboreum* on Alpha Glucosidase, Pancreatic Lipase, and Oxidative Stress; a Bio-Guided Approach

**DOI:** 10.3390/pharmaceutics15112541

**Published:** 2023-10-27

**Authors:** Marwah M. Alfeqy, Seham S. El-Hawary, Ali M. El-Halawany, Mohamed A. Rabeh, Saad A. Alshehri, Aya M. Serry, Heba A. Fahmy, Marwa. I. Ezzat

**Affiliations:** 1Pharmacognosy Department, Faculty of Pharmacy, Modern University for Technology & Information, Cairo 11571, Egypt; heba.fahmy@pharm.mti.edu.eg; 2Pharmacognosy Department, Faculty of Pharmacy, Cairo University, Kasr El Aini, Cairo 11562, Egypt; seham.elhawary@pharma.cu.edu.eg (S.S.E.-H.); ali.elhalawany@pharma.cu.edu.eg (A.M.E.-H.); 3Pharmacognosy Department, College of Pharmacy, King Khalid University, Abha 62251, Saudi Arabia; mrabeh@kku.edu.sa (M.A.R.); salshhri@kku.edu.sa (S.A.A.); 4Pharmaceutical Chemistry Department, Faculty of Pharmacy, Modern University for Technology & Information, Cairo 11571, Egypt; ayaserry@hotmail.com

**Keywords:** *Aeonium arboreum*, metabolic syndrome, antioxidant, α-glucosidase, lipase

## Abstract

Metabolic syndrome (MetS) is a global issue affecting over a billion people, raising the risk of diabetes, cardiovascular disorders, and other ailments. It is often characterized by hypertension, dyslipidemia and/or obesity, and hyperglycemia. Chemical investigation of *Aeonium arboreum* (L.) Webb & Berthel led to the isolation of six compounds, viz. β-sitosterol, β-sitosterol glucoside, myricetin galactoside, quercetin rhamnoside, kaempferol rhamnoside, and myricetin glucoside. Interestingly, *A. arboreum*’s dichloromethane (DCM), 100 and 50% MeOH Diaion fractions and the isolated compound (quercetin-3-rhamnoside) revealed potent α-glucosidase inhibitory activity, especially 50% Diaion fraction. In addition, they also showed very potent antioxidant potential, especially the polar fractions, using DPPH, ABTS, FRAP, ORAC, and metal chelation assays. Notably, the 50% Diaion fraction had the highest antioxidant potential using DPPH and ORAC assays, while the 100% Diaion fraction and quercetin-3-rhamnoside showed the highest activity using ABTS, FRAP, and metal chelation assays. Also, quercetin-3-rhamnoside showed a good docking score of −5.82 kcal/mol in comparison to acarbose. In addition, molecular dynamic stimulation studies illustrated high stability of compound binding to pocket of protein. Such potent activities present *A. arboreum* as a complementary safe approach for the management of diabetes mellitus as well as MetS.

## 1. Introduction

Metabolic syndrome (MetS) is a global issue which raises the risk of diabetes, cardiovascular disorders, and other ailments [1]. MetS affects more than a billion individuals, or around 25% of the world’s population [2]. In fact, it is more prevalent in urban populations in some developing countries than in Western countries [3]. It is often characterized by hypertension, dyslipidemia, obesity, endothelial dysfunction and chronic stress, and hyperglycemia [1]. Hyperglycemia causes vascular damage and endothelial dysfunction, resulting in various vascular complications [4].

On the other hand, obesity is a complex medical condition that means having excessive body fat, which raises the chance of developing other illnesses and conditions such as heart disease, hypertension, diabetes, and some types of cancer [5].

Treatment strategies for MetS primarily depend on dietary and lifestyle modifications, frequently combined with pharmaceutical therapies for MetS-related factors. The most popular treatment strategy for MetS involves lowering postprandial high glucose levels by inhibiting enzymes involved in the digestion of carbohydrates, such as glucosidase and amylase, blocking lipolytic enzymes, such as lipase, inhibiting oxidative stress, and postponing the inflammatory response [6].

The small intestinal epithelium contains an enzyme called α-glucosidase, which is membrane-bound. It facilitates the cleavage of oligosaccharides and disaccharides into glucose, thereby facilitating their uptake into the bloodstream. In fact, acarbose, an α-glucosidase inhibitor, is a promising therapeutic agent for MetS not only because it reduces glucose absorption and postprandial hyperglycemia [1], but also because it improves insulin resistance [7]. Additionally, lipase inhibitors prevent the breakdown of dietary triglycerides into monoglycerides and fatty acids by attaching to the active serine site of intestinal lipase enzymes. Consequently, dietary fat is not absorbed. Such medications are frequently utilized in the treatment of MetS and obesity [8].

Increased production of free oxygen radicals and loss of reduction-oxidation (redox) balance are significant contributors to the pathophysiology of diabetes, hypertension, and ensuing cardiovascular disease. Regular physiological systems depend on reactive oxygen species. On the other hand, loss of redox equilibrium stimulates pro-inflammatory and pro-fibrotic pathways, which hinder insulin metabolic signaling, diminish endothelial-mediated vasorelaxation, and related structural and functional problems in the heart and kidneys. Reversible pro- and anti-free radical reactions are part of the dynamic process known as redox regulation of metabolic function [9].

Antioxidants from different sources must be administered to overcome the complications associated with free radicals. The chemistry behind antioxidant capacity assessments is based on the reactions displayed. These assessments can be divided into two categories: tests based on electron transfer (ET) and tests based on hydrogen atom transfer (HAT) processes. Most HAT-based assays use a competitive reaction strategy in which the breakdown of azo compounds creates competition between antioxidants and substrates for thermally produced peroxyl radicals, as in ORAC (oxygen radical absorbance capacity) and TRAP (total radical trapping antioxidant parameter), while the ability of an antioxidant to reduce an oxidant, which changes color when reduced, is measured via ET-based assays. The concentrations of antioxidants in the sample are associated with the degree of color change as in FRAP (ferric ion reducing antioxidant power), ABTS (2,2′-Azino-bis (3-ethylbenzothiazoline-6-sulfonic acid), and DPPH (2,2-Diphenyl-1-picrylhydrazyl) [10].

Foods made from plants are a natural source of phytochemicals that have been positively linked to the prevention and resolution of MetS clinical symptoms. Plant antioxidants in particular are known to lessen oxidative damage and inflammatory processes connected to obesity and cardiovascular changes [11]. In addition, herbal medications are currently utilized more frequently throughout the world owing to their great efficacy, lack of side effects, and affordable price. In fact, several herbal drugs ameliorate MetS, e.g., Ginsenosides [12], Berberine [13], bitter melon [14], cinnamon [15], and turmeric [16].

The family Crassulaceae, also known as stonecrops or orpines, is taxonomically and morphologically diverse. It has approximately 35 genera, 23 hybrid genera, 1410 species, and 305 intraspecific taxa [17]. Several studies on members of Crassulaceae have demonstrated their antioxidant, anti-hyperglycemic, antihypertensive, anti-inflammatory, antiulcerogenic, antimicrobial, cytotoxic/anticancer, antinociceptive, liver-protective, anti-arthritic, analgesic, anti-malarial, antimutagenic, insecticidal, anti-thrombolytic, and myometrial actions [18]. They are rich in flavonoids, steroids, alkaloids, and triterpenoids [19].

*Aeonium arboreum* (L.) Webb & Berthel belongs to succulents that have a long lengthy tradition of use in various cultures’ ancient medical practices to cure a variety of ailments. In addition to being a diuretic, it has been used for its anti-inflammatory, febrifuge, antipyretic, and anti-hemorrhoidal properties. According to Hassan et al., limited studies investigated the Aeonium genus biologically and chemically [20]. A few studies were conducted on *A. arboreum* (L.) Webb & Berthel and proved its antioxidant, antibacterial and antihypertensive properties [21]. Therefore, the purpose of our study was to explore *A. arboreum* from biological and chemical perspectives. We focused on its potential use in MetS by targeting its α-glucosidase, lipase, and antioxidant effects.

The molecular docking method allows us to characterize how small molecules behave at the binding site of target proteins and to better understand basic biological processes by simulating the interaction between a small molecule and a protein at the atomic level [22]. Prediction of the ligand structure as well as its placement and orientation within these sites (often referred to as pose) and evaluation of the binding affinity are the two fundamental processes in the docking process. Molecular docking was also employed to support the effect of *A. arboreum* as antidiabetic to prepare it or further investigations as preclinical and clinical trials.

## 2. Methods and Materials

### 2.1. Plant Source and Authentication

Non-flowering aerial items of *Aeonium arboreum* were gathered from Saft El Laban, Giza Governorate, Egypt, in December 2018. The specimen was authenticated by Dr. Reem Samir Hamdy, professor at the Faculty of Science, Botany Department, Cairo University, Egypt, and a voucher specimen (Sp. No. 26.6.23) was kept at the herbarium of Faculty of Pharmacy, Cairo University.

### 2.2. Extraction, Fractionation, and Isolation of Major Phytochemicals

Air-dried non-flowering aerial parts of *A. arboreum* (1.5 kg) were grounded and extracted with methanol (MeOH) (4 × 6 L) using Ultraturrax at ambient temperature, up to exhaustion. A 257 g dark greenish-brown residue was obtained from the total extract after it had been filtered and concentrated using a rotary evaporator at 40 °C. The residue was partitioned between dichloromethane (DCM) (4 × 500 mL) after suspension in 500 mL distilled water to give 32.76 g of dried DCM extract besides the mother liquor. The DCM fraction was chromatographed (column A) over SiO_2_ CC (50 × 10 cm, 14 g) using hexane and dichloromethane at a ratio (9:1) as mobile phase to yield 57 fractions. Similar fractions were pooled together according to their TLC (thin layer chromatography) pattern using p-anisaldehyde as spray reagents to visualize the spots on the chromatogram. Fractions (24–26) were combined and re-chromatographed over SiO_2_ CC (50 × 10 cm, 900 mg) using hexane: ethyl acetate at a ratio of (9.5:0.5) as mobile phase to yield 64 sub fractions. Sub-fractions (17–24) were pooled together and further purified to yield compound **1** (17.6 mg), whereas fractions (44 and 45) were pooled together, then chromatographed over SiO_2_ CC (50 × 7 cm, 770 mg) using dichloromethane and methanol in a ratio (9.5:0.5) as mobile phase to yield 50 subfractions. Subfractions (15–25) were combined according to their chromatographic pattern to afford compound **2** (21.8 mg).

A Diaion HP-20 column (100 × 15, 22.5 g), that is a non-polar copolymer styrene-divynilbenzene adsorbent resin, is a polymeric adsorbent used in reverse-phase chromatography. The concentrate of the basic liquor was fractionated and eluted with H_2_O (1 L), followed by MeOH/H_2_O (50%, 2.4 L) then MeOH (100%, 2.7 L) to yield three major fractions A, B, and C (114.3, 3.28, and 1.85 g, respectively).

Fraction C was further chromatographed over SiO_2_ CC (50 × 3 cm, 1.85 g) using ethyl acetate and MeOH gradient elution to obtain 19 fractions. Fraction 7 (3.1 mg) was purified with MeOH using centrifugation and decantation to yield compound **3** (2.9 mg), while fraction 6 was subjected to preparative HPLC (Agilent Technologies, Palo Alto, CA, USA) using acetonitrile and water gradient elution starting from 8:2 to 9:1, respectively yielding compound **4** (2 mg) and compound **5** (1.1 mg). Fraction B was fractionated over a Sephadex LH-20 column (50 × 3 cm, 3.28 g) using H_2_O: MeOH gradient elution to afford 25 fractions. After TLC tracing, fractions 24–25 (30.2 mg) were combined and further purified with MeOH using centrifugation and decantation to yield compound **6** (25.4 mg).

### 2.3. Characterization of Isolated Compounds

The structures of the isolated compounds were elucidated using NMR spectra (Bruker High Performance Digital FT-NMR Spectrometer Avance III (400 MHz)). TMS was used as an internal standard and chemical shifts were given in δ ppm.

### 2.4. Biological Activity

#### 2.4.1. Enzyme Inhibition

##### α-Glucosidase Enzyme Inhibition

*Saccharomyces cerevisiae*’s α-glucosidase was utilized (Sigma-Aldrich, St. Louis, MO, USA Cat#G5003-100UN). Final concentrations of solutions of the specified samples were at 100, 1000 µg/mL, or µM in 5% dimethyl sulfoxide (DMSO) and serially diluted into five concentrations to determine their IC50. Seven concentrations, comprising 31.25, 62.5, 125, 250, 500, 750, and 1000 µM, were prepared from the buffer. This concentration served as the positive control. Using p-nitrophenyl-D-glucopyranoside (pNPG) as a substrate, the inhibitory activity of α-Glucosidase was evaluated colorimetrically by applying the technique described by Gutiérrez-Grijalva et al. with minor modification. In 96-microwell plates, 25 μL of samples/blank were combined with 50 μL of *S. cerevisiae* α-glucosidase (0.6 U/mL) in phosphate buffer (0.1 M, pH 7) for 10 min at 37 °C. The mixture was then incubated once again for 5 min at 37 °C after the addition of 25 μL of 3 mM pNPG as a substrate in phosphate buffer (pH 7). Utilizing a microplate reader (Tecan, Männedorf, Switzerland) to measure the release of p-nitrophenol from the pNPG substrate at 405 nm, enzyme activity was ascertained [23]. After determining the absorbance (A), the inhibition % of -glucosidase was determined using the formula:% Inhibition = [(A blank − A sample)/A blank] × 100

##### Pancreatic Lipase Enzyme Inhibition

The final concentrations of the sample solutions were prepared at 500 µg/mL or 500 µM in 5% DMSO through which the inhibitory concentration 50 (IC50) will be identified. Five concentrations were created by serially diluting the sample concentrations from the initial screening stage that were above 50% inhibition. A stock solution of orlistat (5 µM concentration), the positive control, was prepared in buffer from which 7 concentrations were prepared including 0.039, 0.078, 0.156, 0.313, 0.625, 1.250, and 2.500 µM. P-nitrophenyl dodecanoate (p-NPD from Sigma Aldrich, St. Louis, MO, USA) was used as the substrate, and porcine pancreatic lipase (PPL) was prepared as previously stated by [24] with minor modification. In sum, 25 μL of prepared samples along with a blank were first incubated with 50 L of PPL (1 mg/mL) in a Tris-HCl buffer (100 mM, pH 8) for 10 min prior to detecting the PPL activity. Following the addition of 10 mL of p-NPD (10 mM in isopropanol), the reaction was then incubated again at 37 °C for 20 min. using 200 mL of Tris-HCl buffer as a volume diluent. The percentage of lipase inhibition was calculated using the following equation, using the absorbance recorded at 405 nm on a TECAN microplate reader (Männedorf, Switzerland):% Inhibition = [(A blank − A sample)/A blank] × 100

#### 2.4.2. Antioxidant Activity

##### (2,2-Diphenyl-1-picrylhydrazyl) DPPH Assay

The total methanolic extract and DCM fraction were produced at a concentration of 1 mg/mL in MeOH, while100% and 50% MeOH Diaion Fractions at 0.1 mg/mL in MeOH and the isolated compound (quercetin rhamnoside) at a concentration of 0.4 mg/mL in MeOH. Ascorbic acid was utilized as a positive control, and six concentrations were generated from a main solution (stock) of 100 µM concentration in H_2_O: 7.81, 15.62, 31.25, 62.5, 125, 250, and 500 µM. The process relies on the lessening of the DPPH free radical [23]. Briefly, 20 μL of the sample were placed in a 96-well plate. The solution was then incubated for 20 min in absence of light using 180 μL of the DPPH reagent in 100 μM methanol. Using a microplate reader (Tecan, San Jose, CA, USA) at 540 nm, the reduction in DPPH color intensity that resulted after the incubation period was completed was quantified. According to the following equation, data are presented as means SD:% Inhibition = [(A blank − A sample)/A blank] × 100

Using the linear regression method, the antioxidant impact of the various samples was determined as μM ascorbic acid equivalent.

##### (2,2′-Azino-bis (3-ethylbenzothiazoline-6-sulfonic acid)) ABTS Assay

Samples were made with a 0.1 mg/mL concentration in MeOH, except the total extract and DCM fraction at 1 mg/mL, whereas quercetin rhamnoside was prepared at 0.1 mM. To produce the concentration-response curve, seven concentrations (7.812, 15.625, 31.25, 62.5, 125, 250, and 500 µM), were created from a stock solution of 1000 µM concentration of ascorbic acid, the positive control. The methodology suggested by aitanin, Gomes [25] was used with minor modification to assess the sample’s antioxidant potential to scavenge ABTS free radicals. In brief, the production of the ABTS radical cation (ABTS•+) involved mixing 5 mL of ABTS aqueous solution (7 mM) with 88 L of potassium persulfate (140 mM) while the mixture was left at room temperature for 16 h in the dark. ABTS•+ solution was diluted with methanol to an absorbance of 0.700 (1:50) at 690 nm. Next, 10 µL of sample and 190 µL of ABTS•+ solution were added to a 96-well plate, which was then incubated for 30 min in the dark. The absorbance was then measured at 690 nm using a microplate reader (Tecan, USA). According to the following equation, data are shown as means (*n* = 3) SD:% Inhibition = [(A blank − A sample)/A blank] × 100

Using the linear regression method, the antioxidant effect of the various samples was determined as μM ascorbic acid equivalent.

##### Ferric Reducing Antioxidant Power (FRAP) Assay

The samples were made in a mixture of methanol: DMSO (10:1) at a concentration of 2 mg/mL, while quercetin rhamnoside was made at a concentration of 0.9 mM. To produce the concentration–response curve, seven concentrations, including 7.812, 15.625, 31.25, 62.5, 125, 250, and 500 µM, were created from a stock solution of 1000 µM concentration of ascorbic acid, the positive control. According to the procedure of Benzie et al., this assay was conducted with minor modifications. It depends on Fe^3+^ reduction into Fe^2+^ which forms a colored complex with 2,4,6-tris (2-pyridyl)-s-thiazine (TPTZ). In brief, in a 96-well plate (*n* = 3), 10 mL of the sample was combined with 190 mL of freshly made TPTZ reagent (300 mM acetate buffer (pH = 3.6), 10 mM TPTZ in 40 mM HCl, and 20 mM FeCl_3_, in that order; respectively). The reaction was then allowed to run at room temperature for 30 min while kept in a dark place, and the absorbance was measured at 593 nm. Data are shown as means ± SD [26]. Using the linear regression equation, the antioxidant impact of the various samples was determined as the equivalent of μM ascorbic acid.

##### Oxygen Radical Absorbance Capacity (ORAC) Assay

Trolox (positive control) was prepared at 2 mM in MeOH then serially diluted into ten concentrations 1000, 800, 700, 600, 500, 400, 300, 200, 100, and 50 μM for the concentration–response curve construction. The total extract and DCM fractions were made at 0.2 mg/mL in methanol, 100% and 50% were prepared at 0.05 mg/mL in methanol, while quercetin rhamnoside was prepared at a concentration of 0.225 mM in methanol. The assay was carried out using Liang et al.’s (2014) methodology, with minor modifications. In brief, 10 μL of the prepared samples were allowed to sit at 37 °C for 10 min with 30 mL of fluoresceine (100 nM). For the background measurement, fluorescence measurements (485 EX, 520 EM, nm) were carried out three times for a total of 90 s each (range 1, Figure 1). Then, 70 μL of freshly made 2,2′-Azobis (2-amidinopropane) dihydrochloride (AAPH) (300 mM) was added to each well. For 60 min (40 cycles, each lasting 90 sec), the measurement of fluorescence (485 EX, 520 EM, nm) was maintained [27]. The antioxidant impact of the samples was determined as the μM Trolox equivalent using the linear regression equation, and the data are shown as means (*n* = 3) SD.

##### Metal Chelation Assay

Except for quercetin rhamnoside, all the samples were prepared at a concentration of 0.2 mg/mL in methanol. A stock solution of EDTA (positive control) was made at a concentration of 0.1 mM in water and serially diluted into seven concentrations (80, 70, 60, 50, 40, 20, and 10 μM) for the purpose of creating a concentration–response curve. The method described in [28] was used for the Ferrozine iron metal chelation test, with a few minor adjustments made for microplates. In brief, 50 μL of the sample or substance on a 96-well plate and 50 μL of acetate buffer pH = 6 (*n* = 6) were combined with 20 μL of newly produced ferrous sulphate (0.3 mM in acetate buffer pH = 6). Then, each well received 30 μL of Ferrozine (0.8 mM in acetate buffer, pH = 6). For 10 min, the reaction mixture was incubated at room temperature. At the completion of the incubation period, the FluoStar Omega microplate reader was used to measure the reduction in the intensity of the produced color. The following equation describes how data are expressed as means ± SD:% Inhibition = [(Average Absorbance of blank − Average Absorbance of Test)/Average Absorbance of blank] × 100

The samples’ antioxidant potential was calculated as μM EDTA equivalents using the linear regression equation.

#### 2.4.3. Data Analysis

The data are represented as means ± SD/or SE using Microsoft Excel^®^. Estimation of IC50 was done by Graph-pad-Prism 8^®^. One-way ANOVA was used to analyze the significance of differences between means using Tukey’s test.

### 2.5. Molecular Docking Simulations

Molecular operating environment (MOE) 2019.0102 [29] was used for preparation of both protein and ligand, molecular docking, and evaluation of ligand–protein interaction through visualization of poses and scoring function. Docking was accomplished utilizing the Amber10 procedure and docking placement: Rescoring: London dG, Forcefield, and refinement: Affinity dG; triangular matcher.

#### 2.5.1. Preparation of Target Protein Structure

The PDB id: 2QMJ was used to get human-glucosidase from Protein Data Bank (www.rcsb.org). The MOE’s automatic correction and fixing order allowed for the analysis and resolution of the protein structure. During the protonation stage, hydrogen atoms were introduced to the structure. Following docking completion, findings were observed and filtered using scoring values and visualized poses.

#### 2.5.2. Preparation of Tested Drug Molecules

MOE was used to prepare a 3D model library from the active and selective target compound. The compound in question was put through an energy reduction method and had the partial charges automatically calculated. In order to use it in the docking calculations with the target enzyme, it was ultimately stored as an MDB file.

#### 2.5.3. Validation of Docking

The root mean square deviation (RMSD) is calculated in order to validate the docking process. By redocking the co-crystallized ligand on its target enzyme and then superimposing it on its initial co-crystallized constrained conformation, the RMSD is projected.

### 2.6. Molecular Dynamic Simulations

Based on the docking results, the selected best posed docking complex (Enzyme-quercetin rhamnoside) was subjected to molecular dynamic analysis study. The MD simulations were carried out using the iMod server (iMODS) (http://imods.chaconlab.org/) (accessed on 4 August 2023), which provides a practical interface for this improved normal mode analysis (NMA) methodology in inner coordinates at 300 K constant temperature and 1 atm constant pressure [30]. Finally, a 50 ns molecular dynamic simulation of the target complex was performed.

### 2.7. Swiss-ADME Studies

The Swiss-ADME platform (http://www.swissadme.ch/), (accessed on 3 August 2023) is a freely accessible web tool that gathered the most relevant computational techniques to provide a global estimation of the pharmacokinetics profile of small molecules. Their methodologies were chosen by the web tool creators for their robustness, as well as their ease of interpretation, to enable successful translation to pharmaceutical chemistry. Some of these strategies were updated by online tool designers using open-source algorithms, while others were unaltered versions of the strategies used by the original creators [31]. The molecular structure of the target compound (quercetin rhamnoside) was uploaded into the Swiss-ADME web tool section using simplified molecular-input line-entry specification (SMILES) nomenclature technique using Marvin sketch software 19.19, then the result report was generated.

## 3. Results and Discussion

### 3.1. Isolation of Major Phytochemicals

Six substances were isolated as a consequence of the chemical investigation of *A. arboreum* (Figure 2), which were annotated by comparing their 1H-NMR and 13C-NMR (Appendix A) with those previously reported in literature as well as comparison against authentic compounds on TLC. They were identified as β-sitosterol [19], β-sitosterol glucoside [32], myricetin galactoside [33], quercetin-3-O-α-rhamnoside [34], kaempferol rhamnoside [34], and myricetin glucoside [33]. It is worth noting that this is the first report of β-sitosterol, β-sitosterol glucoside, myricetin galactoside, quercetin rhamnoside, and kaempferol rhamnoside in *A. arboreum* according to our expertise. Previously, *A. lindleyi* was used to isolate β-sitosterol. [19], while myricetin glucoside was detected using LC/MS in *A. arboreum* [21].

### 3.2. Biological Activity

#### 3.2.1. Enzyme Inhibition

##### α-Glucosidase Enzyme Inhibition

The small intestine’s membrane-bound α-glucosidase enzyme, which prevents the breakdown and absorption of carbohydrates, is regarded as a crucial enzyme in the metabolism of carbohydrates. Such enzyme inhibition could lower the high glucose level particularly the postprandial blood glucose and subsequently could be used in management of type II diabetes.

As displayed in Table 1, α-glucosidase was inhibited by acarbose at IC50 of 161.40 µM, all fractions except the MeOH extract revealed more potent inhibitory activity than acarbose, especially 50% Diaion fraction, which was the most potent inhibitor of the enzyme with IC50 of 44.26 µg/mL (Appendix A).

##### Pancreatic Lipase Enzyme Inhibition

Pancreatic lipase enzyme was inhibited by orlistat at an IC50 of 0.70 M, but not until 500 g/mL concentration did any of the studied fractions exhibit significant inhibition as shown in Table 2 and (Appendix A).

#### 3.2.2. Antioxidant Activity

Through scavenging ROS and lowering its potential for destruction, antioxidant activity is essential for optimal physiological functioning of cells. [35]. Utilizing more than two separate approaches allowed for a more precise evaluation of the capacity to scavenge free radicals of various samples [36].

In this research, five separate assays (DPPH, ABTS, FRAP, ORAC, and metal chelation) were performed employing variable techniques for more accurate assessments of the total antioxidant capacity (TAC) [37]; Figure 3 shows the antioxidant capacity of various samples.

In general, preventive antioxidants e.g., superoxide dismutase and catalase are enzyme-based reactions suppressing ROS, while the chain-breaking antioxidants are those which could scavenge tissue ROS, e.g., phenolic compounds and ascorbic acid [37,38]. Their antioxidant mechanisms rely on single electron transfer (SET) and hydrogen atom transfer (HAT), either separately or together. The results of the ABTS assay, which is mainly SET dependent, are confirmed by the DPPH assay, the SET and HAT dependent. Moreover, ABTS can be used for lipophilic or hydrophilic antioxidants [38], whereas FRAP assay is a non-specific redox one which measures the sample’s potential to donate electrons and reduce Fe^3+^-TPTZ complex (colorless) to Fe^2+^-TPTZ complex (blue) in acidic pH [39]. On the other hand, the ORAC assay evaluates the ability of hydrophilic antioxidants to trap peroxyl radicals as a result of HAT mechanism [36]. Furthermore, metal chelation examines the antioxidants’ ability to chelate Fe^2+^ ion in competition with ferrozine through the Fenton reaction.

Interestingly, as illustrated in Figure 3 and in Appendix A), all fractions revealed potent antioxidant capabilities with polar fractions being the most potent, where in DPPH and ORAC Assays, the 50% Diaion fraction showed the highest activity (984.71 ± 93.28 μmolar AAE/mg and 46,781.3 ± 3169.99 μmolar TE/mg, respectively), while in ABTS, FRAP, and the metal chelation assays, the 100% Diaion fraction and quercetin-3-rhamnoside (isolated from 100% fraction) showed the highest activity (1126.8284 ± 87.4056 μmolar AAE/mg and 669.08 ± 33.63 μM TE/mg and 331.12 ± 10.17 μM EDTAE/mg, respectively, for the 100% Diaion fraction) and (1338.1 ± 81.38 μM AAE/mM and 1681.2 ± 49.9 μmolar TE/mM and 377.98 ± 10.17 μmolar EDTAE/mM, respectively, for the quercetin-3-rhamnoside.

### 3.3. Molecular Docking Simulations

From a medicinal perspective, the aforesaid notable activity of quercetin rhamnoside on Saccharomyces α-glucosidase motivated a thorough in-silico study to evaluate the drug’s affinity more fully for human-glucosidase. It should be noted that acarbose has been demonstrated to be much more efficient against human α-glucosidase than against α-glucosidases from other sources, such as Saccharomyces [40]. This top-active drug’s molecular docking affinity against the crystalline human intestinal maltase-glucoamylase α-glucosidase enzyme (hMGAM; PDB ID: 2QMJ) was investigated in comparison to that of the common small molecule inhibitor, acarbose [41]. A monomer-A1 hydrolase glycoprotein complex with various glycans and acarbose that is solved at 1.90 atomic resolution is the implemented 101.02 kDa biological protein target. The co-crystallized ligand acarbose exhibits a fitting across its two initial rings (acarvosine moiety) that is preferred at the hMGAM active-site pocket. Acarbose displayed prolonged polar contacts with the target pocket residues Asp203, Asp327, Arg526, Asp542, and His600 side chains in the X-ray structure. The acarvosine moiety’s hydrogen bonding with Asp443 and Asp571 also included a water-bridge component. The hydrophobic interactions with Tyr299, Ile328, Ile364, Trp406, Trp441, Phe450, Trp539, Phe575, Ala576, Leu577, and Tyr605 give further stabilizing interactions. The 2D interaction pattern of acarbose with hMGAM active site is depicted in Figure 4.

The rigid docking approach was used to redock acarbose into the hMGAM substrate-binding site. According to the results of the acarbose redocking study, the binding score was −24.42 kCal/mol and the RMSD was 0.89 in relation to the co-crystalline ligand, demonstrating superimposition with good fit (Figure 5). These validation results showed that the chosen docking parameters and methods were the best for selecting the ideal docking pose, and the adopted docking protocol was proven to be valid because it showed RMSD values below 2.0 Å.

Table 3 shows a binding score of acarbose as well as the binding interactions with the hMGAM binding site.

A docking score of −5.82 kCal/mol was achieved for quercetin rhamnoside, in contrast, when compared to the co-crystallized hMGAM inhibitor, acarbose. The side chains of numerous important pocket residues, such as Arg202, Asp203, and Lys408, as well as the catalytic Asp542, which is essential for the enzyme’s acid/base catalysis, were seen in the quercetin rhamnoside’s extended hydrogen bond network (Table 3). Similar acarbose-based hydrophobic contact patterns with Phe450, Phe575, and Ala576 offered further stabilizing interactions. Due to its aromatic character, only the docked quercetin rhamnoside demonstrated additional hydrophobic interactions with Trp406 and Met444. Quercetin rhamnoside’s interactions in 2D and 3D with the key amino acids in hMGAM’s binding pocket are shown in Figure 6.

### 3.4. Molecular Dynamic Simulation

Figure 7 shows the outcomes of a molecular dynamics simulation and normal mode analysis (NMA) of the docked complex of human α-glucosidase and quercetin rhamnoside performed using the iMod server (iMODS) (http://imods.chaconlab.org/), (accessed on 4 August 2023). The goal of the simulation study was to ascertain how flexible the target chemical compound was to the α-glucosidase protein residues.

The ability of a specific molecule to distort at each of its residues is determined by the main-chain (protein backbone) deformability. High deformability zones can be used to guess where the chain pivots are. Figure 7A depicts the target protein’s ability to deform after binding with quercetin rhamnoside. In order to improve their ability to bind to the target molecule (quercetin rhamnoside), the amino acid groups have been relocated and are located in the sections of the protein with higher peaks. The B-factor computed from NMA is generated by multiplying the NMA mobility by (8pi2) and the experimental B-factor is acquired from the relevant PDB field. The experimental B-factor is displayed in Figure 7B and is represented as an averaged root mean square deviation (RMSD) in the B-factor column. The enzyme-compound complex was also evident to have a very low eigenvalue of 1.379402 × 10^−4^; the eigenvalue associated with each normal mode denotes the toughness of the motion. The energy needed to deform the structure is directly proportional to the eigenvalue. The lesser the eigenvalue, the easier the distortion (Figure 7C). The covariance map (Figure 7D), whether two residue pairs experience correlated (red), uncorrelated (white), or anti-correlated (blue) motions, is indicated by the covariance matrix. Based on the plausible interactions of the selected compound with hMGAM, we expect that quercetin rhamnoside can serve as a potential drug candidate and target for human α-glucosidase for the treatment of diabetes mellitus.

### 3.5. Swiss-ADME Studies

Swiss-ADME software (http://www.swissadme.ch) accessed on 3 August 2023 was used to calculate the ADME properties and the physicochemical attributes (lipophilicity, size, polarity, solubility, flexibility, and saturation) predictions of quercetin rhamnoside. The target compound seems to have low lipophilic (LogP), soluble and polar molecule (TPSA). It is characterized by low GIT absorption and it cannot pass BBB indicating absence of CNS side effects. It did not exhibit any PAINS structural toxicity alerts. It did not show an inhibitory effect on the metabolic enzymes (CYP2C19, CYP1A2, CYP2C9, CYP2D6, and CYP3A4) revealing absence of hepatotoxicity. It also showed no BRENK structural toxicity alerts. The study showed the necessity for more future modifications for the scaffold so as to improve the ability of oral absorption (Figure 8).

In this study, the purity of the fractions was ensured throughout our research steps starting from proper collection of the desired plant parts and authentication of the plant by an expert (Dr. Reem Samir Hamdy, professor at Faculty of Science, Botany Department, Cairo University, Egypt), as well as proper extraction, fractionation, and isolation using the proper solvents, materials, and procedures as specified. The spectroscopic data of quercetin-3-rhamnoside and other isolated compounds were also presented as Appendix A.

Moreover, the current study addresses the antidiabetic potential of *A. arboreum* for the first time to the best of our knowledge. Interestingly, almost all the fractions of *A. arboreum* showed very potent α-glucosidase inhibitory activity exceeding that of acarbose encouraging further preclinical and clinical studies for its safety and efficacy to be used in the management of diabetes and metabolic syndrome. Other natural products have been traditionally used for their antidiabetic activities by the Chinese, Native Americans, and Indians, e.g., *Rubus parviflorus* and *Amelanchier alnifolia* [42], which need thorough pharmacological and chemical investigations to validate their safety and efficacy. Several efforts have been accomplished to find natural alternatives to ameliorate MetS, e.g., ginsenosides [12], Berberine [13], bitter melon [14], cinnamon [15], and turmeric [16]. Bitter melon (*Momordica charantia*) increases insulin sensitivity but its reported side effects include gastrointestinal symptoms, abortion induction, and depressed fertility [43].

Moreover, the drug–diet interaction of *A. arboreum* should be thoroughly investigated. For example, its administration with ketogenic diet (i.e., a low-carbohydrate and high-fat diet) [44] could lead to severe hypoglycemia; hence, the medication regimen should be properly monitored [45].

Further studies should focus on the incorporation of *A. arboreum* as a promising inhibitor of α-glucosidase activity, in nano formulation for better stability, bioavailability, and efficacy. Moreover, polymeric glucose-responsive systems hold promise for controlling the body’s insulin delivery in response to changes in blood glucose levels and assisting in the maintenance of homeostasis [46].

According to the potent α-glucosidase inhibition results, it will be a promising choice to be used as adjuvant with antidiabetics and to use the plant as a whole due to synergistic effect of constituents as shown in results of assay between total extracts and separate fractions. Finally, a new pharmaceutical candidate was identified for the treatment of metabolic disorders, but further preclinical and clinical studies are required to ensure its efficacy and safety.

## 4. Conclusions

*A. arboreum* is an under-investigated plant with poor studies concerned with its constituents and biological activity. The current phytochemical investigation resulted in the isolation of six major constituents: β-sitosterol (1), β-sitosterol glucoside (2), myricetin galactoside (3), quercetin rhamnoside (4), kaempferol rhamnoside (5), and myricetin glucoside (6). To the best of our understanding, this is the first finding of β-sitosterol, β-sitosterol glucoside, myricetin galactoside, quercetin rhamnoside, and kaempferol rhamnoside in *A. arboreum*.

Different fractions of *A. arboreum* together with quercetin rhamnoside (only one assayed; others not assayed due to lack of amounts for different assays) were assessed for their α-glucosidase, lipase inhibitory, and antioxidant capacities. Interestingly, almost all the fractions showed very potent α-glucosidase inhibitory activity exceeding that of acarbose where the 50% Diaion fraction was the most potent. On the other hand, all fractions revealed very potent antioxidant capabilities, especially the polar fractions are the most, viz. 50%, 100% Diaion fraction and quercetin-3-rhamnoside. Moreover, quercetin-3-rhamnoside showed a good docking score of −5.82 kcal/mol in comparison to acarbose. In addition, molecular dynamic stimulation studies illustrated the high stability of compound binding to pocket of protein. *A. arboreum*’s powerful α-glucosidase and antioxidant activity make them an ideal complement to other safe management strategies for MetS and diabetic mellitus. Nevertheless, further in vivo and clinical studies should be conducted to confirm its effectiveness and safety.

Further studies should now focus on developing and optimizing different dosage forms containing *Aeonium arboreum* standardized extracts and the isolated flavonoid glycosides to be used in the management of metabolic syndrome and diabetes. Moreover, further preclinical and clinical studies are mandatory to ensure their safety and efficacy. Several natural products with antidiabetic and antihyperlipidemic effects have been successfully formulated into different dosage forms; e.g., *Momordica charantia* and *Abelmoschus esculentus* (L.) Moench fruit extracts were formulated as a mixture in a capsule dosage form [47], and Lagerstroemia speciosa leaves’ extract was formulated in soft gel capsules [48]. Others were incorporated into different nano formulations with enhanced bioavailability and efficacy compared to the crude extracts, e.g., micellar nano formulations of *Argyreia pierreana* and *Matelea denticulata* leaves’ extracts [49] and nanostructured lipid carriers of *Leonotis leonurus* leaves’ extract [50].

## Figures and Tables

**Figure 1 pharmaceutics-15-02541-f001:**
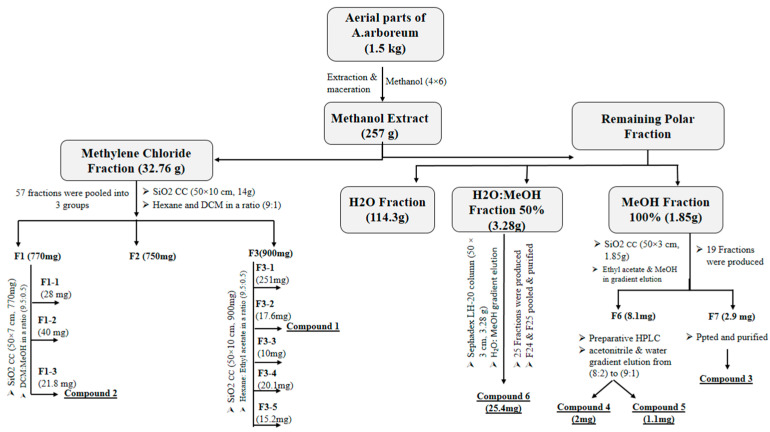
Scheme for *A. arboreum* extraction, fractionation, and isolation of compounds.

**Figure 2 pharmaceutics-15-02541-f002:**
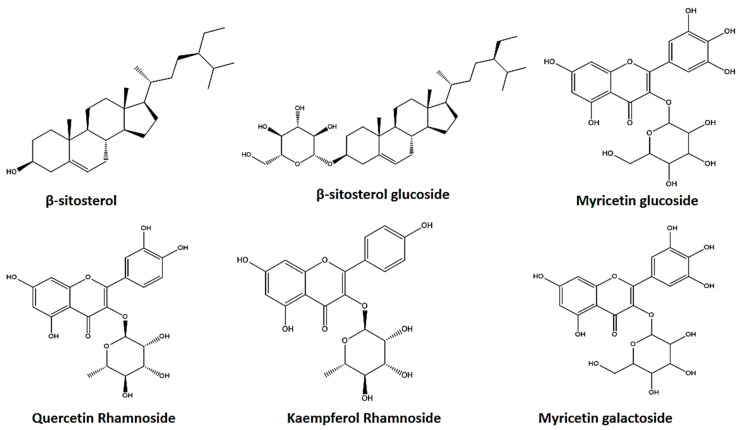
Chemical composition of isolated *A. arboreum* compounds.

**Figure 3 pharmaceutics-15-02541-f003:**
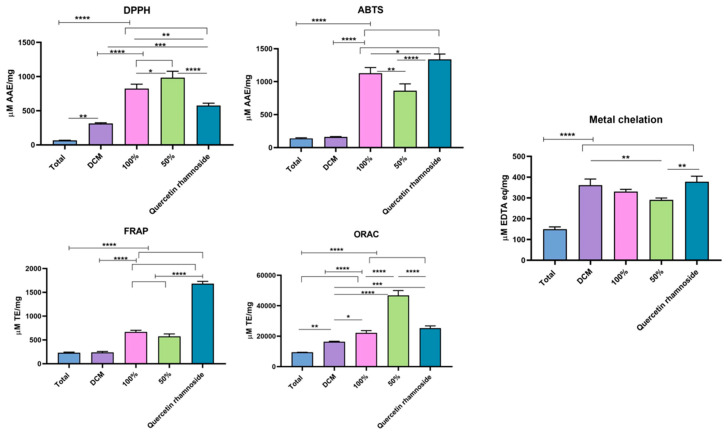
Antioxidant potential (DPPH (2,2-Diphenyl-1-picrylhydrazyl), ABTS (2,2′-Azino-bis (3-ethylbenzothiazoline-6-sulfonic acid), FRAP (Ferric Reducing Antioxidant Power), ORAC (Oxygen Radical Absorbance Capacity), and metal chelation) of plant fractions (Total MeOH extract, DCM (Dichloromethane) Fraction, 100% and 50% Diaion fractions) and quercetin-3-rhamnoside. **** Significantly different at *p* < 0.0001, ***/**/* significantly different at *p* < 0.05.

**Figure 4 pharmaceutics-15-02541-f004:**
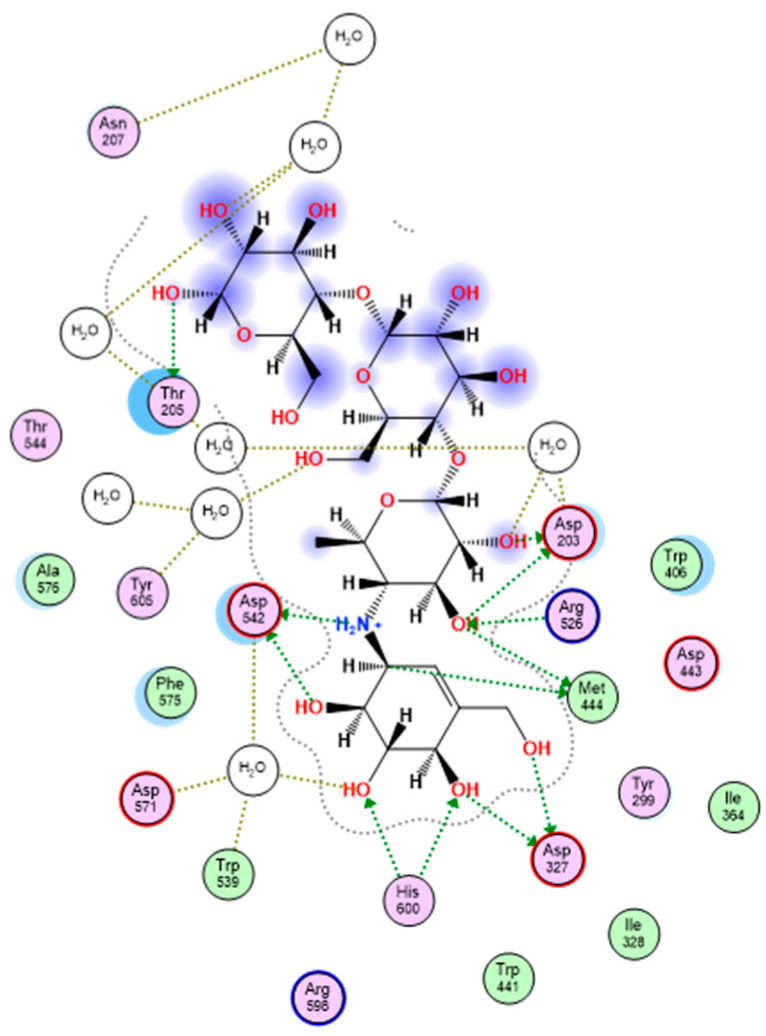
Two-dimensional interactions of acarbose with hMGAM (human intestinal maltase-glucoamylase α-glucosidase enzyme).

**Figure 5 pharmaceutics-15-02541-f005:**
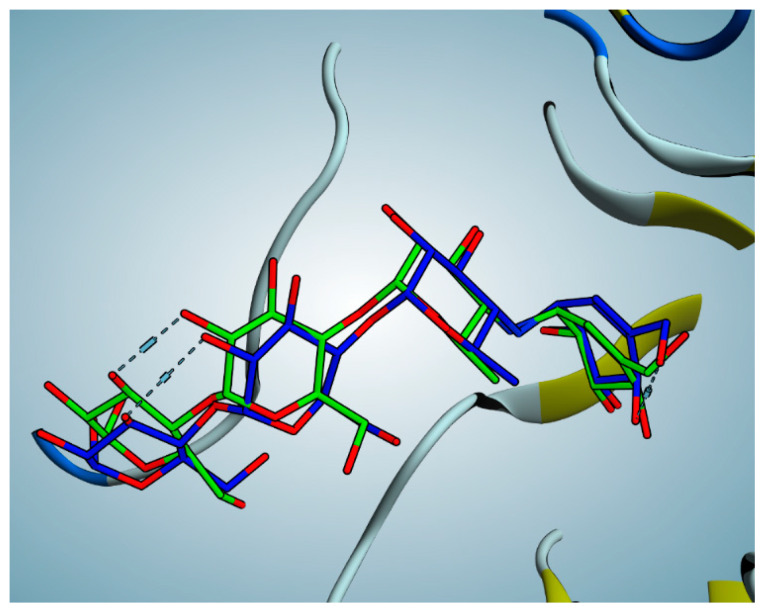
Redocking of acarbose showing perfect superimposition.

**Figure 6 pharmaceutics-15-02541-f006:**
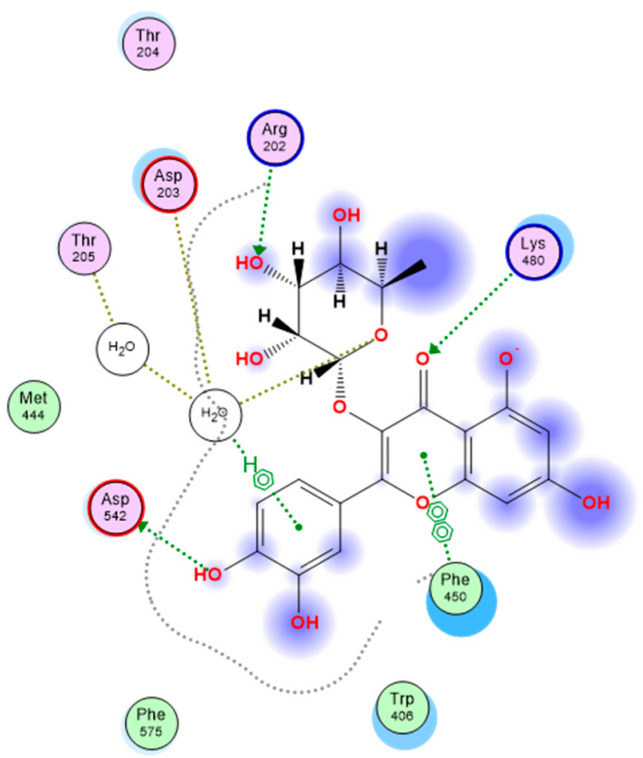
The 2D and 3D interactions of quercetin rhamnoside with the key amino acids of hMGAM (human intestinal maltase-glucoamylase α-glucosidase enzyme).

**Figure 7 pharmaceutics-15-02541-f007:**
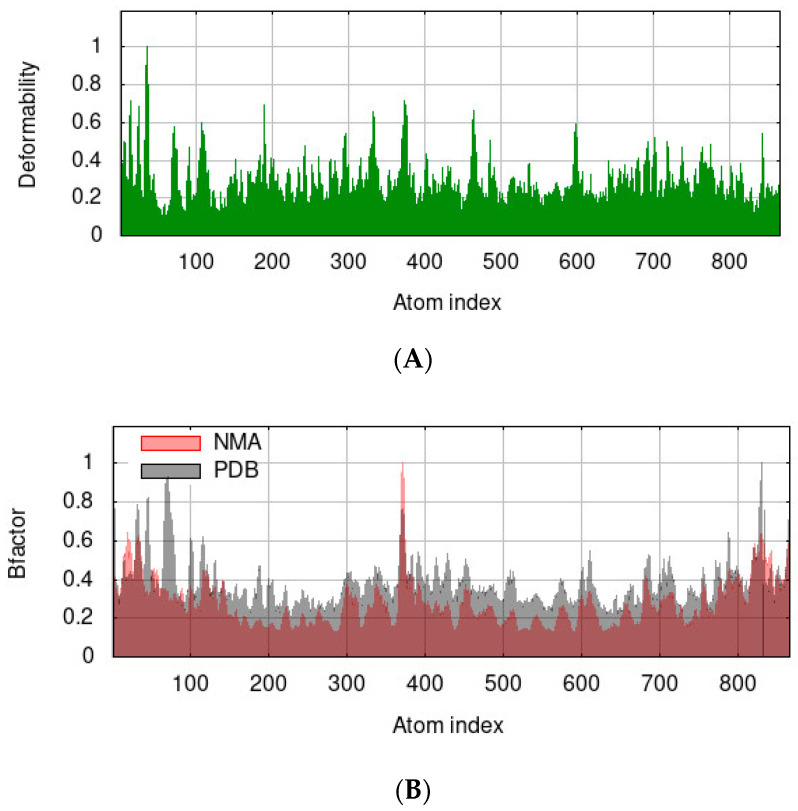
Quercetin rhamnoside/hMGAM complex molecular dynamics simulation via the iMODS server. Deformability, B-factor values, eigenvalues, and covariance model are listed in order from (**A**–**D**).

**Figure 8 pharmaceutics-15-02541-f008:**
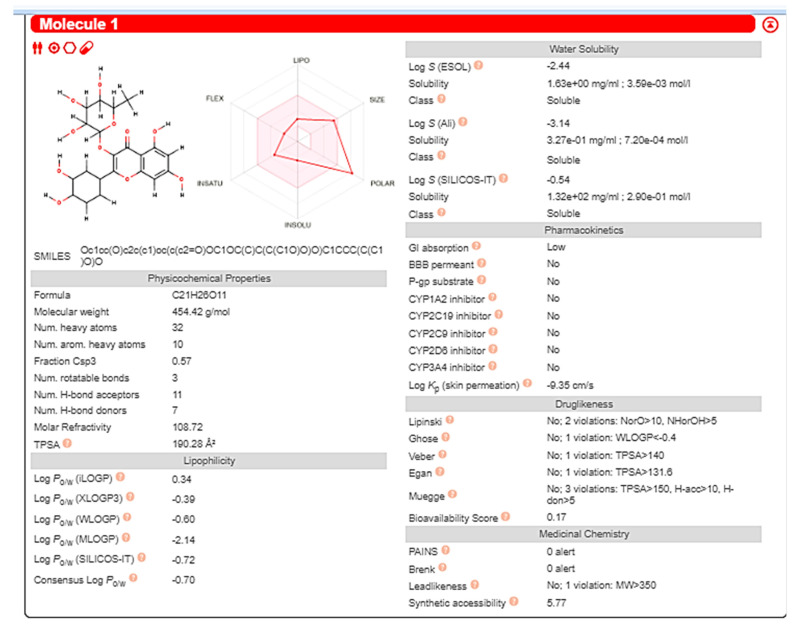
Representation of Swiss-ADME results of quercetin rhamnoside.

**Table 1 pharmaceutics-15-02541-t001:** α-glucosidase inhibitory effect of *A. arboreum* methanol extract, fractions, isolated compound, and acarbose.

Sample	IC_50_ (Mean ± SE)
MeOH extract	>1000 µg/mL
DCM fraction	147.90 ± 1.03 µg/mL
100% MeOH Diaion fraction	48.24 ± 1.02µg/mL
50% MeOH Diaion fraction	44.26 ± 1.06 µg/mL
Quercetin-3-rhamnoside	126.40 ± 1.04 µM
Acarbose	161.40 ± 1.05 µM

**Table 2 pharmaceutics-15-02541-t002:** Pancreatic lipase inhibitory effect of *A. arboreum* methanol extract, fractions, isolated compound, and orlistat.

Sample	% Inhibition (Mean ± SD) (500 µg/mL or µM)	IC_50_ (µM) (Mean ± SE)
MeOH extract	NO	≥500 µg/mL
DCM fraction	23.05 ± 1.05	≥500 µg/mL
100% MeOH Diaion fraction	9.43 ± 0.32	≥500 µg/mL
50% MeOH Diaion fraction	4.88 ± 0.55	≥500 µg/mL
Quercetin-3-rhamnoside	19.06 ± 0.34	≥500 µM

**Table 3 pharmaceutics-15-02541-t003:** Docking score and featured interactions of acarbose against hMGAM enzyme.

	S-Score (kcal/mol)	Bond Length (Å), Involved Receptor Residues	Type of Bond Interaction
Acarbose	−8.66	2.3 Å; His600 (sidechain NH/valienamine 4′-OH)	H-bonding
		2.3 Å; His600 (sidechain NH/valienamine 5′-OH)	
		2.0 Å; Asp203 [sidechain C=O, 6-deoxyglucosyl 4′-OH	
		2.1 Å; 164_; Thr205 (sidechain OH/+3 maltosyl 6′-OH)	
		1.9 Å; Asp203 [sidechain C=O, 6-deoxyglucosyl 3′-OH]	
		1.9 Å; Asp327 (sidechain C=O/valienamine 4′-OH)	
		2.0 Å; Arg526 (sidechain =NHH/valienamine 6′-OH)	
		1.9 Å; Asp542 (sidechain C=O/glycosidic linker NH)	
		Tyr299, Ile328, Ile364, Trp406, Trp441, Phe450, Trp539, Phe575, Ala576	Hydrophobic Interaction

## Data Availability

Data will be available on request from the corresponding author.

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
