# Peer review of "Effect of Phenolics from Aeonium arboreum on Alpha Glucosidase, Pancreatic Lipase, and Oxidative Stress; a Bio-Guided Approach"

_pharmaceutics, 2023, doi:10.3390/pharmaceutics15112541_

Round 1
Reviewer 1 Report
Comments and Suggestions for Authors
This paper describes effects of Aeonium arboretum on enzymatic activities and oxidative parameters in relation with the metabolic syndrome. Although biochemist, I am not an expert in molecular docking and related simulations; this study appears however as well done and clearly described.
Nevertheless, in addition to minor remarks, I wonder if this paper is really in the scope of this Journal. This work is really very “upstream” and its most interesting part is more in the field of biochemistry or medicinal chemistry than “pharmaceutics”. What would be indeed the real pharmaceutical to be developed? Under which form? What would be its “real” – not computer-based – ADME and possible toxicity? … Il believe that all these questions, if not solved – obviously – for this publication should however be evoked.
Other remarks:
· The title should be clarified since it is far too general: most of the study has been performed with quercetin-3-rhamnoside isolated from Aeonium arboretum and focused on α-glucosidase.
· “Diaion HP-20” should be briefly described.
· The purity of the fractions that were used for the biochemical analysis is not specified. This is particularly important for quercetin-3-rhamnoside.
· A scheme of the separation process should be given in the text.
· Pharmacological relevance of the concentrations that were used.
Reviewer 2 Report
Comments and Suggestions for Authors
The article entitled "Aeonium arboreum: an active candidate for inhibiting alpha glucosidase, pancreatic lipase and oxidative stress, a bio-guided approach" is well written and merits publication after minor revision.
- lines 57 - 58 Rephrase the sentence
- More information regarding the approach employed (antioxidant assays & docking part) should be added in the introduction.
Comments on the Quality of English Language
-
Reviewer 3 Report
Comments and Suggestions for Authors
The authors focused on Aeonium arboreum: an active candidate for inhibiting alpha glucosidase, pancreatic lipase and oxidative stress, a bio-guided approach. Please see my suggestions bellow:
Please provide complete information:
- the Model, Producer/manufacturer, City, and Country for EACH APPARATUS (4 information) used in the research, and
- the Producer, Country, purity degree, and concentration or CAS (4 information) used for EACH REAGENT/chemical used. Check the entire manuscript in this regard. This information gives the possibility for replicating you experiment to other authors and are requested in ALL journals.
No needed so many sub-, sub-, subsections. Try merging some of them. Section/subsections of 3-few lines have no relevance as content. They are paragraphs, not sections.
Remove the text L222. It is obvious.
Figures 2, 3, 5. All the Abbreviations used on the figure MUST be explained after its title, below the figure.
Section 3 has the Title Results and discussion, but Discussion are missing!!! Your should do a good Discussion section, developing the ideas below:
· Please describe the potential use of nanotechnology to better improve the delivery of the plants to targeted tissues, considering also the novel/newest molecules in the field – check and consider https://doi.org/10.3390/polym12061397 and https://doi.org/10.3390/ijms24044029
· Discuss the potential interaction of the plant administration with the type of hypo lipidic or hyper lipidic diet. I suggest checking https://doi.org/10.1016/j.lfs.2020.118661
· Also describe the benefits of this plant compared to other plants used for diabetes mellitus such as Momordicia charantia. A figure or table with the beneficial properties specific to this plant would be relevant.
· Detail the potential use of this plants as an adjuvant in diabetes mellitus patients and if it is better to use the plant as a whole or only isolate bioactive principles that can serve as initial stages for drug development.
· Additionally, a last paragraph of Discussion section should be added, describing the Strengths and the Weakness/limitations of your research/results.
Comments on the Quality of English LanguageIt should be revised
Round 2
Reviewer 1 Report
Comments and Suggestions for Authors
The answers provided by the authors are satisfactory and the manuscript has been significantly improved.
Nevertheless, I feel that some of the comments included in the answers of the authors to my criticisms - i.e. description of Diaion HP-20 and purity of the fractions - should be included in the final version of the paper.
Reviewer 3 Report
Comments and Suggestions for Authors
The authors improved their manuscript.
Comments on the Quality of English LanguageMinor errors.
